# How does onchocerciasis-related skin and eye disease in Africa depend on cumulative exposure to infection and mass treatment?

**Natalie V. S. Vinkeles Melchers**[1]*, **Wilma A. Stolk**[1], **Michele E. Murdoch**[2],
**Belén Pedrique**[3], **Marielle Kloek**[1], **Roel Bakker**[1], **Sake J. de Vlas**[1], **Luc E. Coffeng**[1]*

**1** Department of Public Health, Erasmus MC, University Medical Center Rotterdam, Rotterdam, The Netherlands, **2** Department of Dermatology, West Herts Hospitals NHS Trust, Watford General Hospital, Watford, Hertfordshire, United Kingdom, **3** Drugs for Neglected Diseases *initiative* (DND*i*), Geneva, Switzerland

* n.vinkelesmelchers@erasmusmc.nl (NVSVM); l.coffeng@erasmusmc.nl (LEC)

## Abstract

### Background

Onchocerciasis (river-blindness) in Africa is targeted for elimination through mass drug administration (MDA) with ivermectin. Onchocerciasis may cause various types of skin and eye disease. Predicting the impact of MDA on onchocercal morbidity is useful for future policy development. Here, we introduce a new disease module within the established ONCHO-SIM model to predict trends over time in prevalence of onchocercal morbidity.

### Methods

We developed novel generic model concepts for development of symptoms due to cumulative exposure to dead microfilariae, accommodating both reversible (acute) and irreversible (chronic) symptoms. The model was calibrated to reproduce pre-control age patterns and associations between prevalences of infection, eye disease, and various types of skin disease as observed in a large set of population-based studies. We then used the new disease module to predict the impact of MDA on morbidity prevalence over a 30-year time frame for various scenarios.

### Results

ONCHOSIM reproduced observed age-patterns in disease and community-level associations between infection and disease reasonably well. For highly endemic settings with 30 years of annual MDA at 60% coverage, the model predicted a 70% to 89% reduction in prevalence of chronic morbidity. This relative decline was similar with higher MDA coverage and only somewhat higher for settings with lower pre-control endemicity. The decline in prevalence was lowest for mild depigmentation and visual impairment. The prevalence of acute clinical manifestations (severe itch, reactive skin disease) declined by 95% to 100% after 30 years of annual MDA, regardless of pre-control endemicity.

**Data Availability Statement:** All relevant data are within the manuscript and its Supporting Information files.

**Funding:** NVSVM, WAS, BP, and LEC received funding (grant no. AID-OAA-G14-00010) from United States Agency for International Development (USAID, www.usaid.gov) through the Drugs for Neglected Diseases initiative (DNDi, www.dndi.org). The contents are the responsibility of the authors and do not necessarily reflect the views of USAID or the United States Government. WAS, SJdV, and LEC also gratefully acknowledge funding of the NTD Modelling Consortium(www.ntdmodelling.org) by the Bill & Melinda Gates Foundation (www.gatesfoundation.org) through grant OPP1184344. In addition, LEC was financially supported through an Innovational Research Incentives Scheme Veni by the Dutch Research Council (NWO, www.nwo.nl/en) via ZonMw (www.zonmw.nl/en, grant 016.Veni.178.023). The multi-country study on pre-control onchocercal skin disease by MM received financial support from the UNDP/World Bank/WHO Special Programme for Research and Training in Tropical Diseases (TDR, tdr.who.int). The funders had no role in study design, data collection and analysis, decision to publish, or preparation of the manuscript.

**Competing interests:** The authors have declared that no competing interests exist.

## Conclusion

We present generic model concepts for predicting trends in acute and chronic symptoms due to history of exposure to parasitic worm infections, and apply this to onchocerciasis. Our predictions suggest that onchocercal morbidity, in particular chronic manifestations, will remain a public health concern in many epidemiological settings in Africa, even after 30 years of MDA.

## Author summary

Onchocerciasis, also known as river blindness, is the second most common infectious cause of blindness worldwide, but also leads to serious skin conditions. Large-scale interventions are ongoing to control and eliminate the disease in Africa, yet the impact of these interventions on onchocercal morbidity is largely unknown. Here, we predict the trends in a wide spectrum of skin and eye disease due to onchocerciasis after up to 30 years of annual mass drug administration (MDA) with ivermectin. To this end, we have developed a novel disease framework within the established ONCHOSIM model. We show that annual MDA will rapidly reduce the prevalence of acute clinical conditions, whereas the prevalence of chronic clinical manifestations will decline much more slowly. The new disease framework was validated with several data sources and reproduced morbidity trends adequately, making the framework applicable for more refined disease prevalence predictions by taking account of treatment history in Africa. Such predictions are essential for accurate estimates of disability-adjusted life years lost due to onchocerciasis by 2025.

## Introduction

*Onchocerca volvulus* is a parasitic filarial nematode transmitted through the bite of infected blackflies (genus *Simulium*). In endemic areas, individuals may build up considerable worm loads through life-long exposure to bites in the absence of treatment [1]. Adult worms reside in worm bundles located in palpable subcutaneous nodules or in deeper body tissues, and produce microfilariae (mf) that migrate throughout the body, mainly to the skin and eyes [2]. Adult female worms live for 10 years on average [3] and produce hundreds to thousands of mf daily. Clinical manifestations are triggered, among others, by the host immune response to the release of both microfilarial antigens and endosymbiotic *Wolbachia* bacteria when mf die, and by the resulting tissue damage [4–7]. Clinical manifestations caused by inflammation are diverse, including onchocercal skin disease (OSD) and onchocercal eye disease (OED). OSD can be very severe and includes deforming skin lesions and itching. The accumulation of tissue damage can eventually lead to irreversible stigmatising skin pathologies, i.e. depigmentation (leopard skin), hanging groin, and atrophy [8]. Mf-induced damage to the eye can lead to visual impairment and eventually blindness. Blindness in turn may lead to premature death [9–11]. Approximately 218 million people in 30 countries worldwide (2018) are at risk of onchocerciasis; 99% of those people live in sub-Saharan Africa [12]. According to estimates, about 7.5 million people were infected with *O. volvulus* in West-Africa around 1974 (prior the implementation of the Onchocerciasis Control Programme [OCP]) [13]. Another study estimated that 36 million people would have been infected in the APOC countries by 2011 if there had been no ivermectin treatment [14].

To deal with the dramatic health and associated socio-economic impact of onchocerciasis, large-scale control programmes based on vector control and/or preventive therapy to control onchocerciasis in Africa have been running since 1974. Mass drug administration (MDA) with ivermectin decelerates *O. volvulus* transmission by killing the larval stage parasites (mf) in humans, and by temporarily interrupting and permanently reducing mf production by adult female worms [15]. It has been suggested that repeated ivermectin treatments may also have a macrofilaricidal effect on adult worms, especially when individuals are treated at high frequency (≥4x/year) [16–18]. Studies in foci in Mali, Senegal, and Nigeria demonstrated that the prevalence of skin mf can be reduced below postulated threshold values for elimination using ivermectin treatment only [19–22]. These achievements have led in 2010 to an expansion of the original World Health Organization (WHO) objectives for morbidity control to include elimination of onchocerciasis transmission [23].

Monitoring and evaluation has hitherto largely focussed on MDA coverage and its effect on *O. volvulus* infection. However, the underlying goal remains reduction in morbidity and it would also be useful to identify to what extent interventions have reduced disease prevalence, what the disease burden is at present, and what it will be in the future. Mathematical models have previously been used to predict the impact of interventions on *O. volvulus* infections and disease [24–27]. Although there are modelling studies on the predicted impact of MDA in terms of infection [28], severe itching, and eye disease [29], there are, to date, no estimates for the whole spectrum of onchocercal morbidity. To predict the prevalence of onchocerciasis-related clinical manifestations (i.e. severe itch, reactive skin disease, palpable nodules, depigmentation, atrophy, hanging groin, visual impairment, and blindness) over time, we extended the established individual-based transmission model ONCHOSIM [24–26,30] with a novel module for the development and natural history of morbidity. We have used this new disease module to predict how the prevalence of onchocercal skin and eye morbidity decline during MDA, in order to assess the expected remaining prevalence after up to 30 years of MDA.

## Methods

### The simulation model ONCHOSIM

ONCHOSIM is an established individual-based mathematical model for the transmission and control of onchocerciasis in a dynamic population [24,30,31]. A detailed formal description of the ONCHOSIM model including the Java source code is provided elsewhere (see additional files in [26]). Previous versions of ONCHOSIM included a basic disease process that only accommodated chronic, irreversible clinical manifestations, and could simulate one condition at a time. Here, we report findings with ONCHOSIM 2.76, a version which incorporates a new module for morbidity to simulate a wide spectrum of onchocercal skin and eye disease simultaneously. S1 Text provides a detailed description of the structure and quantification of ONCHOSIM; S1 Text also contains all supplementary tables and figures, meaning that, for instance, "S1 Table" refers to "S1 Table within S1 Text".

### Generic disease module

The new, generic disease module within ONCHOSIM can simulate a wide range of clinical manifestations due to onchocerciasis (Table 1), which can be reversible (severe itch, reactive skin disease) or irreversible (depigmentation, atrophy, hanging groin, visual impairment, and blindness). Tissue damage is caused by the host immune response to the release of both microfilarial antigens and endosymbiotic *Wolbachia* bacteria when mf die and that induce inflammatory reactions (skin and eye manifestations). In the model, the amount of tissue damage changes over time as new tissue damage accumulates with every time step (in months) due to

**Table 1. List of clinical manifestations that are modelled in ONCHOSIM, and the associated assumptions about reversibility.** The input specifications and disease parameters for each clinical manifestation are presented in S3 Table within S1 Text.

| Disease process | Clinical manifestation | Reversibility of condition |
|---|---|---|
| Reactive skin disease (acute and chronic papular onchodermatitis & lichenified onchodermatitis) | Any reactive skin disease (RSD) | Reversible |
| Severe itch | Severe itch (itch with insomnia) | Reversible |
| Depigmentation | Threshold 1: mild depigmentation | Irreversible |
| | Threshold 2: severe depigmentation | |
| Atrophy | Atrophy | Irreversible |
| Hanging groin | Hanging groin | Irreversible |
| Onchocercal eye disease | Threshold 1: visual impairment | Irreversible |
| | Threshold 2: blindness | |

dying mf (see S1 Table for mathematical equations and explanation of how tissue damage is simulated). Acute, reversible clinical manifestations may to some extent disappear through a constant healing process, defined as a constant fraction of damage that is healed with every time step (damage regression rate). For irreversible clinical conditions, we assume zero regression of tissue damage.

Clinical manifestations are assumed to appear when an individual passes a critical threshold of accumulated tissue damage. For irreversible conditions, these are considered to be permanent but reversible clinical manifestations resolve once accumulated tissue damage drops below the threshold. Clinical conditions with a disease continuum (i.e. mild to severe depigmentation and visual impairment to blindness) are governed by the same counter of accumulated tissue damage. Here, clinical manifestations can develop in a two-phase process based on separate disease thresholds (threshold 1 and 2 in Table 1), with a higher disease threshold for the more severe form of the clinical condition. For visual impairment and blindness, the threshold is assumed to differ between forest and savanna bioclimate, i.e. lower for savanna, to reflect the generally higher prevalence of eye disease. Mf killed through ivermectin treatment are not considered to directly cause tissue damage, but ivermectin reduces the mf load, and can thereby temporarily halt the accrual of tissue damage. There are some reports of adverse effects (generic symptoms, e.g. oedema, fever, pain) upon ivermectin treatment within 24–48 hours after intake, but these reactions were generally mild and self-limiting [32–34]. There is evidence that *O. volvulus* mf in patients treated with ivermectin first migrate to regional lymph nodes where they degenerate and are encircled by eosinophils or macrophages. As a result, inflammatory cellular reactions due to the death of mf in the tissues upon ivermectin intake is minimal [35] (in contrast to diethylcarbamazine that gives a strong histological reaction within ocular tissue accelerating onchocercal blindness [36]). Treatment is further assumed to only indirectly affect the development and presence of symptoms via removal of mf which would have caused damage if they would have died naturally.

We further incorporated some degree of variation in susceptibility to specific clinical manifestations between hosts by varying the amount of damage accrued per dying mf, using a random life-long susceptibility index for each person and clinical manifestation. There is evidence that there are various determinants that lead to variation in susceptibility to infectious disease susceptibility, including host and pathogen genetic variation and immune effectiveness [37,38]. As a result, some individuals will develop a particular clinical condition very rapidly, and others will develop it slowly or never, for a given adult worm and mf load. We assume that different types of conditions (e.g. skin disease and eye disease) may develop independently within the same host. This means that an individual may be more prone to develop one particular symptom (e.g. severe itch) than another (e.g. eye disease).

Finally, we accounted for excess mortality due to blindness. As in previous modelling exercises [39], ONCHOSIM models the excess mortality by reducing the individual's remaining life expectancy once he/she becomes blind by a user-defined fraction (usually 50%) [39]. With this assumption, the model adequately reproduced the declining trend in blindness during vector control, shown by data from the Onchocerciasis Control Programme (OCP) across West-Africa, assuming that blindness is irreversible [10,40]. The excess mortality due to blindness also slightly affects the presence of other clinical manifestations, as these are now modelled simultaneously and aggregate in those with the highest worm burdens.

The parameters of the disease module (S3 Table) have been quantified by fitting the model against data for pre-control age patterns of disease prevalence, pre-control association of infection and disease prevalence, and longitudinal trends of the effect of MDA on disease prevalence. More details about case definitions and data of onchocercal skin and eye disease are given below.

## Modelling development of palpable nodules as clinical condition

Palpable nodules due to the presence of patent female worms in an individual are a proxy for infection at the population-level, but nodules can also be considered a clinical manifestation that exert a disease burden due to shame and stigmatisation. The presence of adult worms is recognised by the human body as foreign material, and leads to thickened epidermal cell layers, i.e. palpable collagenous nodules. We have, therefore, followed a similar approach as for the clinical manifestations in Table 1, but now assuming that disease development is triggered by the presence of adult patent female worms. Again, we accounted for individual variation in susceptibility to developing palpable nodules. On average, adult worms have a long lifespan of approximately 10 years [3]. When adult worms eventually die without replacement, the process leading to nodule formation will cease and the palpable nodules may disappear over time [41].

## Onchocercal skin disease (OSD)

**Case definitions.**  Case definitions for each morbidity subtype were made according to the classification of Murdoch *et al*. 1993 [8]. We combined acute papular onchodermatitis, chronic papular onchodermatitis, and lichenified onchodermatitis as one clinical manifestation: reactive skin disease (RSD). Depigmentation was assumed to be a multi-stage skin disease in this model, progressing from incomplete pigment loss to complete pigment loss with spots of normally pigmented skin, or 'leopard skin' [8]. Here, we refer to those two stages as mild and severe depigmentation (Table 1).

**Data.**  To quantify model parameters for OSD, we used anonymised individual-level data on multiple clinical manifestations for 6,910 individuals from five African countries (Cameroon, Ghana, Nigeria, Tanzania, and Uganda) [42]. The data contain clinical information on severe itch, reactive skin disease, palpable nodules, mild and severe depigmentation, atrophy, and hanging groin by age and sex. The only indicator of infection available in the data was the presence of palpable nodules. We restricted our analyses to data from Nigeria, Tanzania, and Uganda, including a total of 4,810 randomly sampled persons from 24 villages. Data from Cameroon and Ghana were excluded in view of potential bias introduced by convenience sampling.

**Stratification by endemicity for OSD.**  The data were stratified into three endemicity classes based on pre-control nodule prevalence in adult males (aged $\geq$20 years): mesoendemic ($\geq$ 20% and <40%), hyperendemic ($\geq$40% and <65%), and very hyperendemic ($\geq$65%) villages (S1 Text, section 2.1). We converted the prevalence of palpable nodules (as a proxy of

infection in the data) into mf prevalence in order to reproduce pre-control associations of infection and morbidity. To do this, we took the mean prevalence of palpable nodules for each of the three endemicity categories (meso-, hyper, and very hyperendemic), and translated it into a mean mf prevalence on the basis of a previously published function for converting nodule prevalence in adult males to OCP-standardised (mean) mf prevalence in the general population aged ≥5 years [43]. This conversion reflects the average nodule and mf prevalence of an entire region; it does not consider the level of uncertainty associated with village-level prevalence. We then tuned the relative biting rate (rbr) values in the model such that with a high number of repeated simulations, our model could adequately reproduce the mean mf prevalence per endemicity category. See S1 Text, section 2.1 for more information.

### Onchocercal eye disease (OED)

**Case definitions.**   Here, we use the term "visual impairment" for any moderate or severe visual impairment. Following the WHO criteria, we defined visual impairment as visual acuity between 6/18 and 6/60 and equal to or better than 3/60 in the better eye. According to the WHO criteria, we defined blindness as visual acuity of less than 3/60 or a restriction of visual field to less than 10˚ in the better eye [44,45].

**Data.**   We used pre-control data on the association between the community-level prevalence of infection and the prevalence of visual impairment and/or blindness to quantify our model. We quantified OED for forest and savanna bioclimates separately, since the savanna strain is more pathogenic, resulting in different biological outcomes for the different parasite species [46]. Data from the savanna bioclimate reported mf prevalence in the population aged ≥5 old [47], whereas the data from forest and mixed savanna-forest bioclimate consisted of community microfilarial load (CMFL) as the infection proxy [48–57]. CMFL is a measure of intensity of infection in the community; it is defined as the geometric mean number of mf per skin snip among adults aged 20 years and more [41].

**Infection intensity for OED.**   To reproduce the association between infection and OED at the community-level, we defined a large number of rbr values (from 0.280 to 0.980), resulting in a range of pre-control infection levels that covered the range of the data. This was then used to relate model-predicted OED prevalence with mf prevalence (savanna) or CMFL (forest).

### Calibration of parameters and validation of the model

Basically, there are three free parameters for each clinical manifestation: variation in an individual's susceptibility to damage, disease threshold, and rate of damage regression. For reversible clinical manifestations (i.e. severe itch, RSD, palpable nodules), we first chose a grid of values for the damage regression rate. For each chosen value of regression rate, we calibrated parameters for variation in individual's susceptibility to damage and disease threshold, using data on pre-control association between age patterns and prevalence of disease for the different age groups and endemicity strata [42]. Then, based on the fit to the available longitudinal data of the impact of six years of MDA on the prevalence of reversible clinical manifestations [58,59] (S1 Text, section 2.1), we chose which the optimal combination of (chosen) damage regression rate and (fitted) values of variation in individual susceptibility and damage threshold.

For irreversible clinical conditions, only two parameters needed to be estimated, as the damage regression rate was considered to be zero. For OED, as well as irreversible subtypes of OSD (i.e. mild and severe depigmentation, atrophy, hanging groin), we fitted the variation in individual's susceptibility to damage and disease threshold such that the model could best reproduce the observed pre-control association between age patterns and prevalence of disease

for the different endemicity strata simultaneously [42] (for subtypes of OSD), and such that the model could best reproduce the pre-control association between infection intensity and prevalence of OED. The disease threshold for OED in forest areas was fitted using pre-control data on blindness and visual impairment combined. No data were available on the prevalence of visual impairment for savanna areas, but there is evidence from hyperendemic OCP-savanna areas that the pre-control prevalence of visual impairment is about 1.8 times the prevalence of blindness [39,48]. Additionally, the mean mf prevalence in a hyperendemic area has been reported to be 73% [39], so we modelled the prevalence of visual impairment for savanna areas as 1.8 times the prevalence of blindness at 73% mf prevalence. The assumption of zero damage regression was supported by a Cochrane review of placebo-controlled trials that found no evidence for an effect of ivermectin on severe eye disease [60].

Disease parameters related to each clinical manifestation were quantified using a two or three-dimensional grid search of the difference between model predictions and actual empirical age-specific morbidity prevalence for each endemicity category, expressed by the sum of squared errors (SSE). Further details of the SSE grid search are described in S1 Text, section 2.3.

After quantification of the different disease parameters for each clinical manifestation, we validated the disease model post-hoc using internal and external data. Amongst others, we simulated the model-predicted ecological association between the prevalence of infection (here: palpable nodules) and skin morbidity at the community-level, and assessed how well this fitted pre-control field data [42,58,59,61]. Details of the model validation are presented in S1 Text, section 3.

## Predicting trends in morbidity during MDA

We ran simulations for various scenarios to evaluate the impact of MDA on onchocercal morbidity over time pertaining to pre-control endemicity, bioclimate (for OED), and history of MDA (annual *vs.* semi-annual, therapeutic coverage of 60%, 70% and 80%). The prevalence of infection and disease in hypoendemic areas was taken as a 0.10 fraction of that of mesoendemic areas, as in previous work [39]. We modelled MDA for a duration of 30 years, i.e. the maximum number of treatment rounds for any MDA implementation unit ("project") of the African Programme for Onchocerciasis Control (APOC). To assess how long it takes for the various disease outcomes to largely disappear from a population, we estimate after how many years of MDA the prevalence of clinical manifestations falls below an arbitrary threshold of 0.5%. We note that this may be longer than the duration of MDA required for interruption of transmission, as the latter does not require infection to be completely cleared from a population [25,62] and because chronic symptoms like blindness persist after clearing infection [63]. For each scenario, we present the average of 750 repeated simulation runs, as some of the disease outcomes were quite rare. For simulating scenarios, we used the rbr values that reproduced the mean pre-control mf prevalence as reported by Prost *et al.* [64] for meso-, hyper-, and very hyperendemic areas.

## Sensitivity analyses

We performed multiple univariate sensitivity analyses, including alternative MDA therapeutic coverages ranging between 60% and 80%, annual versus semi-annual MDA with 70% therapeutic coverage, 1% and 5% systematic non-participation of the population eligible to take ivermectin during MDA, a 1% regression of OED before the disease threshold has been reached, and between 40% and 60% reductions in the remaining life expectancy for blind individuals. For the latter two sensitivity analyses, we re-quantified the disease parameters (i.e.

variation in individual's susceptibility to damage and disease threshold) to reproduce the pre-control data.

## Results

### Parameter estimates and goodness-of-fit to data

The pre-control model-predicted prevalence patterns for each of the subtypes of skin manifestations fitted reasonably well with the data (Fig 1). The disease thresholds for severe itch (255) and RSD (210) were quite similar (yet with different individual variation in susceptibility to disease and regression rates, S3 Table), resulting in similar age patterns (Fig 1). The disease thresholds for atrophy (11.3 thousand) and hanging groin (21.4 thousand) were much higher than those of the acute clinical manifestations and depigmentation (mild: 2.4 thousand; severe: 4.3 thousand). The disease threshold for hanging groin was almost twice as high as for atrophy (but with substantially lower individual variation in susceptibility to disease), reflecting that hanging groin is a much rarer clinical manifestation. This very high disease threshold for hanging groin corresponds with the very low prevalence (<3%) of hanging groin in 50+ year old individuals as compared to atrophy (<5%) from ≥30 years old (Fig 1). The disease threshold of palpable nodules (triggered by adult patent female worms, not mf) equals 12.

The model-predicted pre-control association between prevalence of blindness and OED (i.e. sum of visual impairment and blindness) against the prevalence of skin mf in the population of ≥5 years for savanna and CMFL in forest areas also followed the observed data adequately (Fig 2). The difference between the disease thresholds of visual impairment and blindness was higher in forest areas (10.5 vs. 12.5 thousand, respectively) than in savanna areas (1.7 vs. 3.1 thousand, respectively), which reflects that only a small additional amount of tissue damage for people with visual impairment in savanna areas is needed to become blind (S3 Table).

### Validation of the model with external data

Our model performs reasonably well when validating the ecological association of our pre-control model predictions against internal and external data (Fig 3). As expected, the model-predicted association of the prevalence of nodules in adult males with the prevalence of morbidity closely follows the data of Murdoch *et al*. 2002 [42] (data used for fit). When comparing the model-predictions with external data, the prevalence of atrophy and hanging groin was underestimated by our model as compared to the Kaduna dataset. This underestimation can largely be explained by the higher reported pre-control prevalence rates of these subtypes of OSD in Kaduna, Nigeria [61]. There was also a discrepancy between the model-predicted prevalence of itch as compared to the data from Kaduna, which is due to the fact that we quantified our model solely with data on the prevalence of severe itch whereas Murdoch *et al*. 2017 [61] included the prevalence of troublesome itch in their clinical survey. They defined troublesome itch as any form of itching with or without insomnia, whereas severe itching is defined as itching with insomnia [42].

Likewise, when assessing the age-stratified prevalence of subtypes of OSD using the Kaduna data, our model underestimates the prevalence of nodules in older age groups (>35 years), atrophy (from ≳30 years), and hanging groin (from ≳40 years). There seems to be a lower prevalence of RSD in Kaduna, Nigeria as compared to the prevalence reported by Murdoch *et al*. 2002 [42] (S5 Fig).

The model-predicted concurrence of clinical manifestations fitted the data reasonably well (S6 and S7 Figs). Trends in the model-predicted prevalence of morbidity over time since the start of MDA also matched the observed data quite well (S8 Fig). Although data for any

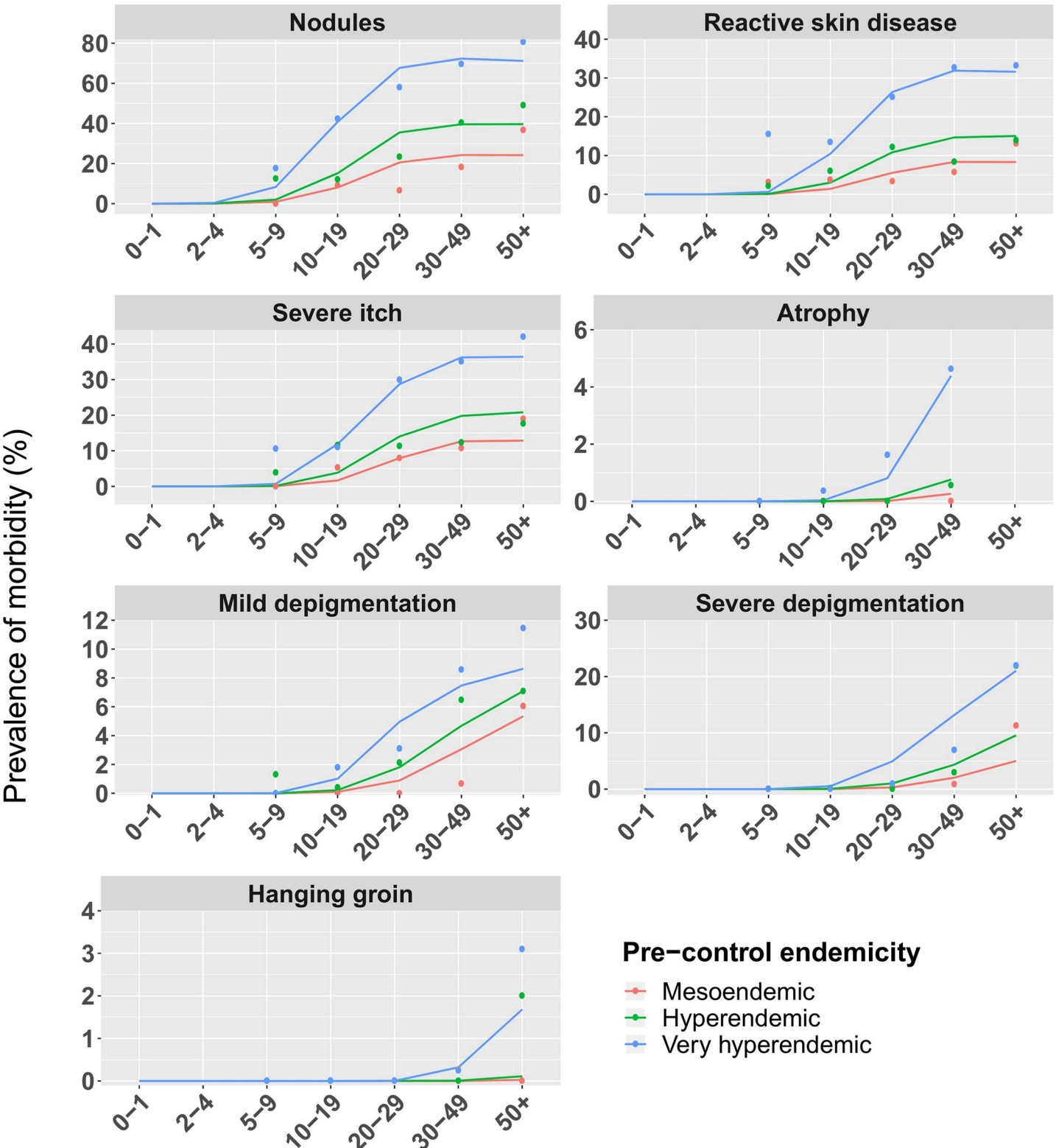

**Fig 1. Comparison of ONCHOSIM model predictions with multi-country data by age patterns for different onchocercal skin disease subtypes and endemicity strata.** The data points [42] consist of three onchocerciasis endemic countries with individual data from 4,810 persons (≥5 years) living in 26 villages. Each panel represents a different disease manifestation or disease stage. The lines represent the model predictions by ONCHOSIM by pre-control age patterns in prevalence of onchocercal skin disease. Please note the different scales for the y-axes in the panels.

depigmentation before and during control deviated from our model predictions [59], the model-predicted pattern of the decline in prevalence of depigmentation over time since the start of MDA was similar to the external data (i.e. a slightly decreasing straight line, meaning a very slow decline in morbidity prevalence).

## Model-predicted impact of MDA on the prevalence of disease

Fig 4 shows the model-predicted reduction in prevalence of morbidity after multiple years of annual MDA with different population coverages. The corresponding impact on prevalence of infection over time is shown in S17 Fig. To reduce the prevalence of palpable nodules to <0.5% in mesoendemic communities, between 15 and 20 years of annual MDA are required, depending on the therapeutic coverage of MDA achieved. In very hyperendemic areas, the predicted prevalence of palpable nodules will still be approximately 14% after 30 years of annual

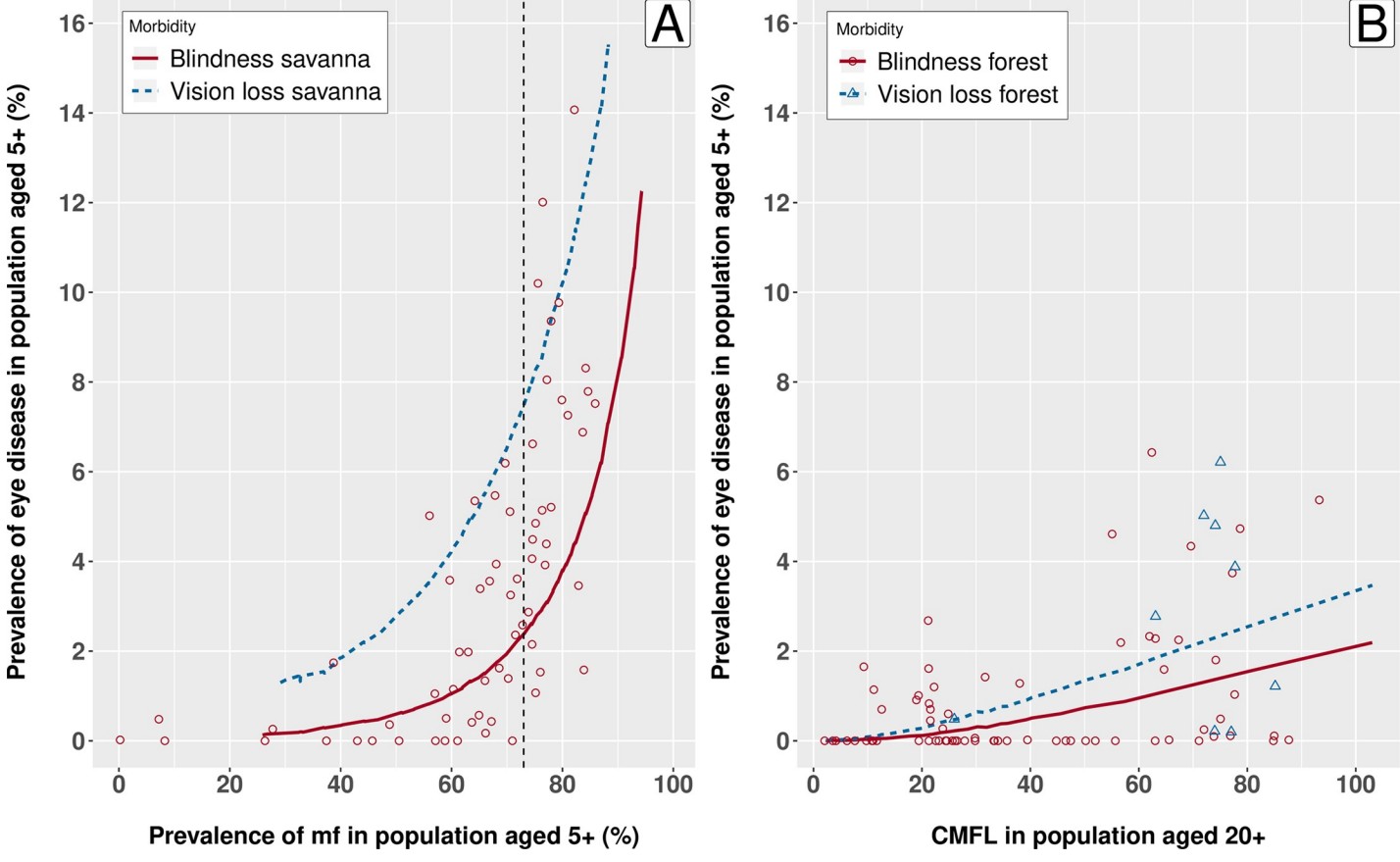

**Fig 2. Pre-control prevalence of onchocercal eye disease (OED) by mf prevalence in the total population: model predictions versus observed data (data points) for A) savanna areas, and B) forest areas.** The red and blue lines represent the model-predicted pre-control prevalence of blindness and vision loss (any OED, i.e. blindness + visual impairment), respectively. In panel 2A the vertical dotted line is the mf prevalence at 73% at which the prevalence of visual impairment is assumed to be 1.8 times the prevalence of blindness [48]. In panel 2B the blue triangles represent data points for the data on any OED [39]. Please note the different infection metrics for measurement of onchocerciasis prevalence (mf prevalence versus community microfilarial load, CMFL).

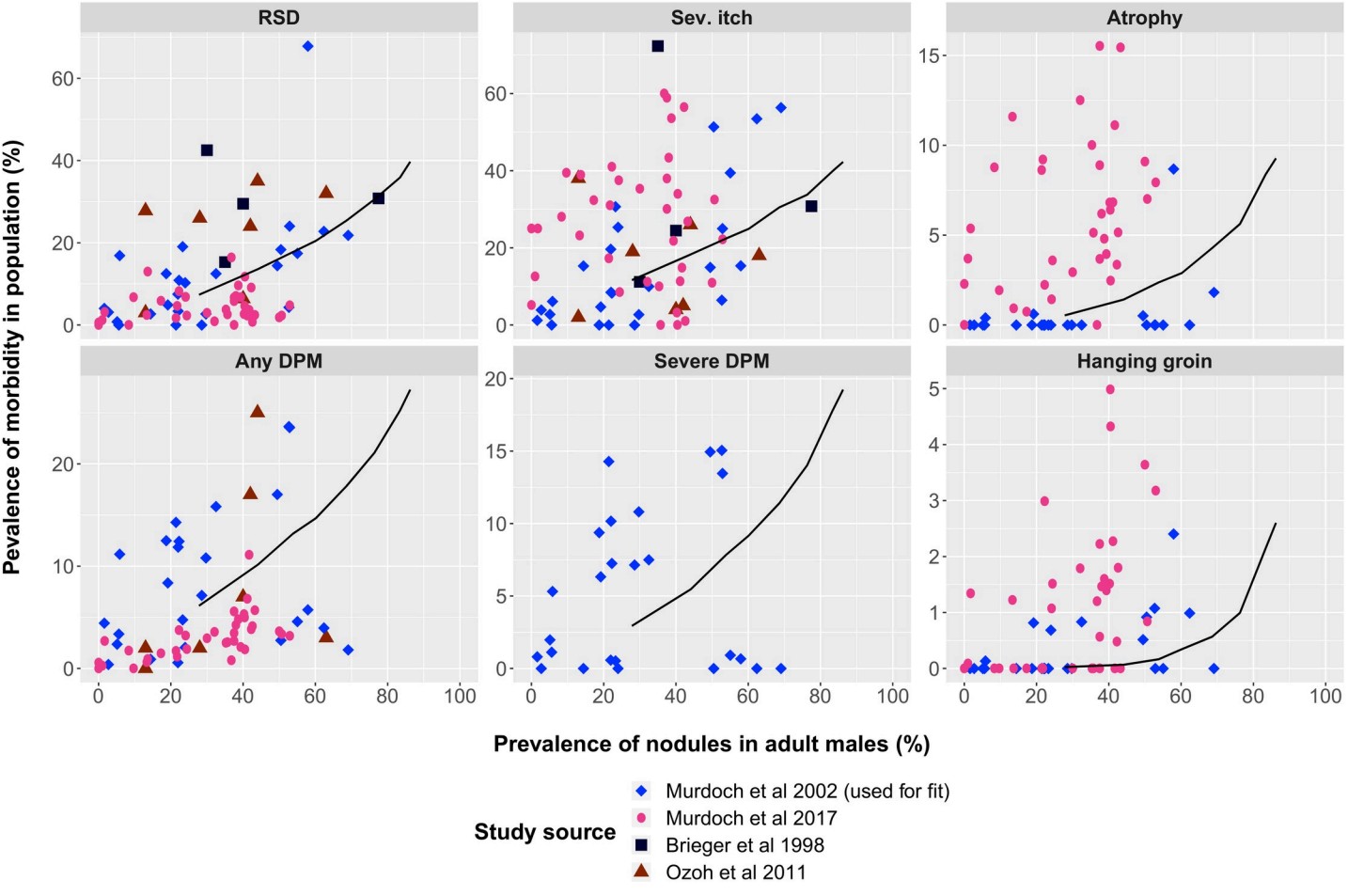

**Fig 3. The model-predicted association between prevalence of nodules and prevalence of subtypes of OSD at the community level.** The data [42,58,59,61] consist of a combined 19,781 persons living in 64 villages in Africa. Data were collected by means of cross-sectional dermatological surveys of individuals aged ≥5 years. Each panel represents the prevalence of a different disease manifestation or disease stage (y-axis). The black lines represent the pre-control model predictions by ONCHOSIM for prevalence of onchocercal skin disease by prevalence of infection. The model-predictions for mf prevalence rates below 15% are not shown due to unstable runs. Please note the different scales for the y-axes in the panels. The data from Murdoch *et al.* 2002 [42] (blue coloured bullets) were used for the quantification of the model, and the remaining data sources [58,59,61] were used for external validation of the model. The data from Murdoch *et al.* 2017 [61] only contains data on the prevalence of troublesome itch rather than severe itch. Troublesome itch was defined as any form of itching with or without insomnia, whereas severe itching was defined as itching with insomnia.

MDA (75% relative reduction since pre-control; Table 2) with an average MDA coverage of 60%. The prevalence of nodules can be further reduced by increasing the coverage to 70% (about 4% prevalence after 30 years of annual MDA; 93% relative reduction) or 80% (<0.5% prevalence after 30 years of annual MDA; 99% relative reduction).

As reversible clinical manifestations correlate more to current infection status than to history of infection, when MDA is implemented we see a faster prevalence reduction for severe itch and RSD than for irreversible conditions. This is readily explained by the fact that during MDA, prevalence of acute symptoms declines simultaneously in all age groups (S9 Fig). In mesoendemic areas, RSD could be reduced to <0.1% after 11 years of annual MDA at 70% coverage, but in very hyperendemic areas this is expected to take 28 years. The reduction in prevalence of RSD to <0.5% in very hyperendemic areas could be achieved more rapidly with 15 rounds of annual MDA by increasing the population coverage to 80%. These numbers are slightly less optimistic for the prevalence of severe itch, where in mesoendemic areas a predicted prevalence of <0.5% can be reached after a minimum of 15 years of annual MDA, even

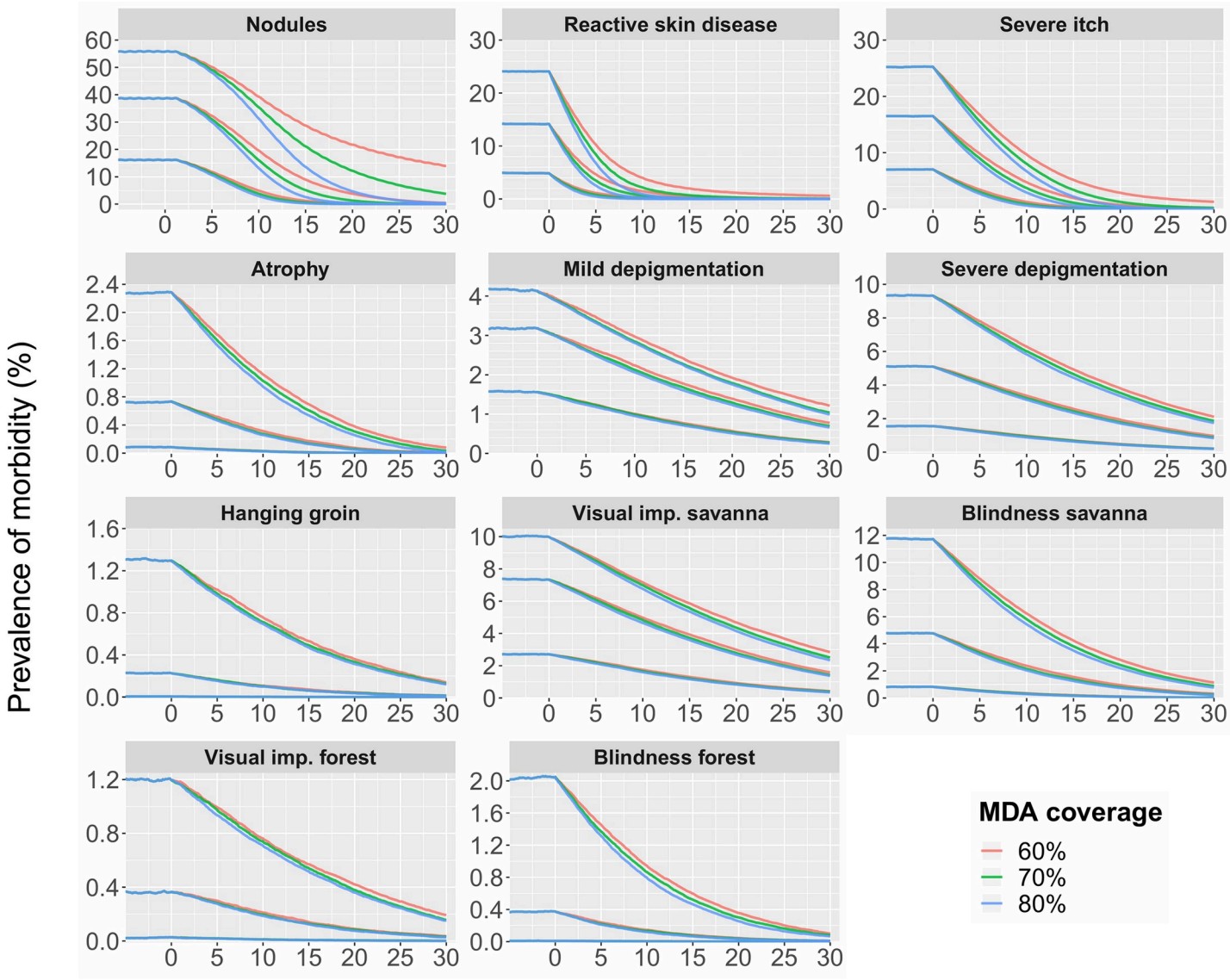

**Fig 4. Predicted impact of annual mass drug administration (MDA) on the prevalence of morbidity due to onchocerciasis.** Coloured lines represent different levels of treatment coverage. The lines that start at different points on the y-axis represent various pre-control endemicity levels (from upper to lower lines: very hyperendemic, hyperendemic, mesoendemic). Different panels represent the various subtypes of onchocercal skin disease (OSD) and onchocercal eye disease (OSD). The predicted trends are based on the average of 750 simulations. Please note the different scales for the y-axes in the panels.

with 60% MDA coverage. In the worst-case scenario (very hyperendemic areas pre-control and 60% MDA coverage), it would require over 30 rounds of annual MDA to reach ~1.5% prevalence. Still, this is a 95% relative reduction in prevalence since pre-control over a 30-year time frame.

More annual rounds of MDA will be required for irreversible clinical manifestations than for reversible conditions, since the reduction in prevalence is slower and more linear for the former. This is readily explained by the fact that during MDA, the decline in prevalence of chronic symptoms is mostly driven by demographic turn-over, as can be seen from the shift in

**Table 2. Model-predicted relative reduction in morbidity prevalence in the general population after 30 years of annual MDA.** RSD = reactive skin disease; DPM = depigmentation.

| Pre-control endemicity and MDA coverage | Onchocercal skin disease | | | | | | | Onchocercal eye disease | | | |
|---|---|---|---|---|---|---|---|---|---|---|---|
| | Nodules | RSD | Severe itch | Depigmentation | | Atrophy | Hanging groin | Visual impairment | | Blindness | |
| | | | | Mild | Severe | | | Forest | Savanna | Forest | Savanna |
| **Very hyperendemic settings** | | | | | | | | | | | |
| 60% | 100 | 100 | >99 | 80 | 84 | 87 | 89 | 84 | 72 | 95 | 90 |
| 70% | 100 | 100 | 100 | 79 | 84 | 87 | 89 | 87 | 75 | 96 | 93 |
| 80% | 100 | 100 | 100 | 79 | 85 | 88 | 91 | 88 | 77 | 97 | 94 |
| **Hyperendemic settings** | | | | | | | | | | | |
| 60% | 100 | 100 | 100 | 84 | 88 | 90 | 92 | 90 | 79 | 98 | 93 |
| 70% | 100 | 100 | 100 | 84 | 87 | 91 | 90 | 92 | 81 | 98 | 95 |
| 80% | 100 | 100 | 100 | 84 | 89 | 92 | 92 | 92 | 82 | 98 | 96 |
| **Meso- and hypoendemic settings** | | | | | | | | | | | |
| 60% | 100 | 100 | 100 | 87 | 89 | 93 | 93 | 89 | 84 | 100 | 97 |
| 70% | 100 | 100 | 100 | 86 | 90 | 93 | 93 | 92 | 86 | 100 | 97 |
| 80% | 100 | 100 | 100 | 87 | 91 | 93 | 98 | 95 | 87 | 100 | 98 |

age patterns in S9 and S10 Figs. Yet, for hanging groin, atrophy, and OED in forest areas, less than 30 years of annual MDA are required to reduce these conditions to <0.5% prevalence, thanks to their very low initial pre-control prevalence levels. For example, in meso- and hyper-endemic areas, the pre-control prevalence of atrophy is already below 1%, and annual MDA will assist slowly in removing this clinical condition from these communities. In very hyperen-demic areas, the pre-control prevalence of atrophy is 2.3% and it will take an average of 15 to 20 years of annual MDA to reduce its community prevalence below 0.5% (~70% relative reduction in prevalence since pre-control). These patterns in reduction of prevalence are simi-lar for hanging groin and OED in forest areas.

For the more common depigmentation and OED in savanna areas, more rounds of annual MDA are required to reduce morbidity in communities to low levels. In mesoendemic areas, the minimum duration of annual MDA to reduce the prevalence of mild and severe depigmen-tation to <0.5% would be between 20 and 25 years with 60% MDA coverage (~64% relative reduction in prevalence compared to pre-control). Increasing the MDA coverage to 80% would only marginally increase the relative reduction in prevalence to ~67%; the clinical con-dition will only slowly fade from the population. The pre-control prevalence of mild and severe depigmentation (4.2% and 9.3% respectively) is much higher across very hyperendemic areas as compared to areas of moderate endemicity (1.6% and 1.5% respectively). Due to the higher pre-control prevalence levels, we predict that more than 30 years of annual MDA are required to reduce the prevalence of depigmentation to <0.5%.

For visual impairment in very hyperendemic savanna areas–even though the pre-control prevalence of blindness (11.8%) in these areas is higher than visual impairment (10.0%)–more than 30 annual rounds MDA will be required to reach <0.5% prevalence, even with 80% popula-tion coverage (similar to depigmentation). Reducing the prevalence of savanna visual impairment to very low levels will require more MDA rounds than the number of rounds required for savanna blindness. This is because the lower disease threshold for visual impairment still allows some new cases to develop over time (although this likelihood is reduced with continued annual MDA), but it is highly unlikely that these individuals will become blind.

The stochastic variation of the model for the scenario of annual MDA with 70% treatment coverage is presented in S11-S13 Figs. For relatively low pre-control disease prevalences (<10%, i.e. atrophy, mild depigmentation, hanging groin, and OED in forest areas), there is

somewhat more stochastic variation in individual runs, meaning that prediction uncertainty is higher for the number of annual MDA rounds to reduce morbidity prevalences to <0.5%.

## Sensitivity analysis

Increasing the MDA frequency from annual to biannual will result in a more rapid decline in prevalence of infection (S18 Fig) and reversible clinical conditions compared to pre-control (S19 Figs). For example, in very hyperendemic areas, semi-annual MDA led to <0.5% prevalence of RSD and severe itch about seven years earlier than with annual MDA only, whereas semi-annual MDA almost halved the time to reach <0.5% prevalence of palpable nodules as opposed to annual MDA. These programmatic differences only slightly impact the speed of the reduction of the prevalence of irreversible clinical conditions since the implementation of MDA, as these are mostly driven by demographic turn-over. The assumption of higher systematic non-participation to MDA barely impacts any of our results (S22-S24 Figs).

Assuming 1% instead of 0% reversibility of tissue damage leading to vision loss changed the curvature of the pre-control association between community infection levels and the prevalence of OED (S14 Fig). This change was caused by a shift in the estimated damage threshold for blindness that compensated for the fact that people (who are yet to turn blind) are constantly recovering from eye damage. Likewise, alternative assumptions about excess mortality due to blindness (i.e. 40% and 60%, instead of 50%) changed the curvature of the pre-control association between infection levels and OED (S15-S16 Figs). Here, the shift in the estimated damage threshold for blindness compensated for the change in remaining lifespan of prevalent blind cases. In addition, the alternative assumptions about excess mortality due to blindness also influenced the prevalence of concurrent chronic skin conditions. As a result, assumptions about reversibility of eye damage and excess mortality due to blindness together influenced the prevalence of reductions in eye and skin morbidity over time after implementation of MDA (S25-S29 Figs), particularly for OED, but also rare subtypes of OSD (such as hanging groin). More information on the quantification of these biological assumptions and their impact on morbidity is described in sections 5.2 and 5.3 of S1 Text.

## Discussion

We have developed and quantified a new disease module within the established mathematical model ONCHOSIM to evaluate the impact of MDA on morbidity prevalence with projections up to 30 years. We quantified the model using a wide variety of robust empirical data on the pre-control association of infection and onchocercal skin and eye disease from various African countries. We also used longitudinal trends to quantify and validate the model, and found that observed disease patterns could be reproduced adequately. In areas of very high pre-control onchocerciasis endemicity, the relative reduction in the prevalence of chronic morbidity ranged from 70% to 89% after 30 years of moderate annual MDA coverage (60%). The prevalence of acute clinical manifestations (severe itch, RSD) declined to almost zero after 30 years of annual MDA, independent of the pre-control endemicity and MDA coverage. However, the speed of this decline depends on the pre-control endemicity, MDA coverage, and frequency of MDA rounds.

This is the first time that multiple clinical manifestations due to onchocerciasis have been simultaneously simulated in a mathematical model that is able to differentiate between reversible and irreversible clinical manifestations, as well as single- and multi-stage disease, taking account of excess mortality due to blindness in the trends for prevalence of all these conditions. For OED, we quantified the occurrence of morbidity separately for savanna and forest areas to reflect that levels of OED are generally higher in savanna than forest areas with

moderate parasite burdens [65,66]. For the quantification of the reversible clinical skin manifestations, we also used longitudinal trends. For longitudinal impact of ivermectin on itch, we used the Brieger et al. [58] study, one of the few studies on the impact of ivermectin on itch that distinguishes severe itch (which is more specific for onchocerciasis). For the remaining morbidity patterns, we included longitudinal data collected by the same investigators using the same screening methods to limit heterogeneity between studies [59]. In addition, our model predictions agree well with reported impact of eight to ten years of annual MDA on palpable nodules in selected hyperendemic villages in Nigeria, Cameroon and Uganda [67,68].

Our model predictions for itch are not directly comparable to earlier estimates derived with a previous version of ONCHOSIM [39] or EpiOncho [27]: we quantified our model only for "severe itch" (itch with insomnia) as, following expert opinion, this was considered more specific for *O. volvulus* infection than the definition of "troublesome itch" used in previous modelling. Several factors contribute to differences in the model-predicted trends in itch prevalence during MDA. Firstly, the earlier estimates were based on a simple statistical relationship between the prevalence of troublesome itch and adult female worms [27,39], while we now consider the dynamic accumulation and regression of tissue damage. Secondly, as discussed in detail elsewhere [69], differences in underlying transmission dynamics make ONCHOSIM more optimistic about elimination prospects than EpiOncho, which explains why the number treatment years needed to reduce itch prevalence to low levels is much lower in the current analysis as compared to the previous estimate from Turner *et al.* [27]. As our model-predicted prevalence of reversible clinical manifestations (severe itch, RSD, and palpable nodules) closely followed longitudinal data on the impact of MDA [58,59], we are confident that our estimates are robust.

As acute, reversible clinical manifestations are directly correlated to active *O. volvulus* infection, intensified MDA effectively reduces the prevalence of these subtypes of OSD more rapidly than irreversible conditions. Although intensified MDA also leads to a more rapid decline in incidence of chronic forms of OSD and OED, this barely influences the rate of decline in prevalence. The prevalence of irreversible clinical manifestations diminishes gradually over a longer timeframe through a natural process of gradual mortality in the affected population and an influx of healthy people through birth in the absence of new cases. Still, this process is slow and a substantial number of chronic cases is expected to remain after 30 years of MDA. New therapeutic drugs may target populations affected by onchocerciasis more effectively and further prevent the development of new clinical signs. For instance, moxidectin treatment causes a longer sustained reduction in individual skin mf densities than observed with ivermectin treatment [70]. Still, better treatment cannot reduce the existing chronic burden of disease.

At the time of the quantification of the model, we did not yet have the pre-control data from Kaduna (savanna area, Nigeria) [61] to our disposal. We therefore used these data as an external data source to cross-validate our model. As there is significant variability in the characterisation of cutaneous signs, such as of depigmentation, as well as potential variability between study designs or screening approaches, we have only used data where the cutaneous signs are defined according to Murdoch et al [8], for both the model quantification as well as the external validation. The variability in methodological approaches as well as differences in bioclimatic and epidemiological settings, may lead to different associations between infection and disease. For example, in communities of southern Cameroon, a higher prevalence of any depigmentation was reported for a given nodule prevalence as compared to our association, i.e. for a nodule prevalence of 40%, they reported a ~30% prevalence of any depigmentation, and at 80% nodule prevalence a ~60% depigmentation prevalence [71]. On the other hand, the associations between the prevalence of infection and depigmentation in rural villages in Kwara

State in Nigeria [72] and the Republic of Congo [73] are of the same order of magnitude as our predictions. The prevalences of chronic papular onchodermatitis and lichenified onchodermatitis (included within the category RSD in our analysis) as well as palpable nodules were all lower in savanna-Kaduna (Nigeria) [61] as compared to the data from forest and mixed forest-savanna areas used in this study [42]. On the other hand, the prevalence of hanging groin and atrophy were higher in the savanna communities. Such geographical variation may occur by chance, although we cannot exclude the possibility of systematic differences in the prevalence of subtypes of OSD between savannah and forest areas, similar to variations measured in OED prevalence related to genetic variation in *O. volvulus* [46,65,66,74]. For endemic areas previously under the APOC mandate, the difference in the occurrence of skin disease between forest versus savanna areas is of less relevance as the majority of endemic areas are of the forest type. However, when implementing the model in other areas (e.g., countries formerly covered by OCP), one should be aware of potential differences in morbidity prevalence between the bioclimates. Although we used the most robust data published on the association between infection and OSD in forest areas [42], there might still be some uncertainty and unexplained variation between regions and countries in the prevalence of subtypes of morbidity.

We quantified the prevalence of OED assuming irreversibility of clinical manifestations and 50% excess mortality due to blindness. Changing these biological assumptions influenced the disease threshold estimates and thereby the pre-control shape of the association between the prevalence of infection and morbidity substantially. Evidence for the reversibility of OED is weak, and most studies assessing the impact of ivermectin on the reduction of OED are underpowered or of too short duration [60,75–77]. On the one hand, community-based studies reported a decline in the prevalence of early-stage eye lesions (i.e. punctate keratitis, iritis) after several rounds of ivermectin intake [78,79]. This may suggest (partial) reversibility of early-stage OED. On the other hand, no reductions in the incidence or prevalence of more severe OED—such as sclerosing keratitis, chorioretinitis, and optic atrophy—were measured after two years of semi-annual community-wide MDA [79]. This may suggest irreversibility of more severe onchocercal eye damage. It is therefore difficult to assess whether our assumptions on (ir)reversibility of OED are realistic. The 50% reduction in remaining life expectancy (excess mortality) used in the baseline analyses was previously predicted [39] using data from OCP on trends in blindness during vector control in villages with a pre-control mf prevalence of 70% and 90% [40]. Although this reduction may be a biologically reasonable assumption, the excess mortality rates due to (all-cause) blindness may systematically vary between countries, bioclimates, populations, and time periods [10,11,80]. Likewise, women may be more prone to a shortened life expectancy due to blindness than men [10,80]. These variations in excess mortality rates due to (all-cause) blindness between settings also introduces uncertainty in our estimates of total mortality due to blindness.

One point that we have not considered here, but which might need to be taken into account in future studies if evidence becomes available, is the possibility of excess mortality due to microfilarial load in an individual [81,82] or severe subtypes of OSD (i.e. severe itch, hanging groin, atrophy). We have also not considered onchocerciasis-associated epilepsy (OAE). In view of the growing evidence [83,84], it might be interesting to also include this disease manifestation in our model to allow for disease predictions over time. Finally, we have used dynamic mechanistic processes of disease accumulation to quantify our model. Such a mechanistic mathematical model is more appropriate to forecast disease prevalence over time than statistical modelling. However, fitting such more sophisticated stochastic models with advanced techniques to quantify uncertainty (e.g. Bayesian frameworks) is technically highly challenging and computationally demanding, and we therefore opted for a simpler approach for model quantification and instead performed extensive sensitivity analyses.

## Conclusion

We have developed, quantified, and validated a new disease module within ONCHOSIM to model trends over time of the prevalence of onchocercal skin and eye morbidity since the implementation of MDA. Our model has shown for the first time how the prevalence of various manifestations of onchocerciasis are likely to decrease with ongoing MDA with ivermectin. It is anticipated that with future input from a wider field of clinicians, including those with expertise in onchocerciasis-associated epilepsy, further refinements to the model may be developed. We expect that chronic onchocerciasis morbidity will remain a significant public health problem now and in the near future. This disease module will be used to estimate trends in the onchocercal disease burden of in terms of Disability Adjusted Life Years lost due to onchocerciasis in Africa with stratifications by age, sex, and country.

## Ethics approval

Not applicable. Anonymous secondary data were used and approval for their use was provided.

## Supporting information

**S1 Text. Formal mathematical description of the model, parameter values used for the predictions, annotated input and output files, as well as more detailed methods of the quantification, model validation, simulation, and results of the sensitivity analyses.**
(PDF)

## Acknowledgments

We thank Drs. MC Asuzu, M Hagan, WH Makunde, P Ngoumou (deceased), KF Ogbuagu, D Okello, G Ozoh, WR Brieger and JHF Remme for their contributions to the multi-country pre-control onchocercal skin disease study and for sharing the raw data set. We would like to warmly thank Louise Burrows (DND*i)* for help with editing the final manuscript.

## Author Contributions

**Conceptualization:** Wilma A. Stolk, Michele E. Murdoch, Belén Pedrique, Sake J. de Vlas, Luc E. Coffeng.

**Data curation:** Natalie V. S. Vinkeles Melchers, Marielle Kloek.

**Formal analysis:** Natalie V. S. Vinkeles Melchers, Luc E. Coffeng.

**Funding acquisition:** Luc E. Coffeng.

**Investigation:** Natalie V. S. Vinkeles Melchers, Luc E. Coffeng.

**Methodology:** Luc E. Coffeng.

**Project administration:** Luc E. Coffeng.

**Software:** Roel Bakker, Luc E. Coffeng.

**Supervision:** Wilma A. Stolk, Sake J. de Vlas, Luc E. Coffeng.

**Visualization:** Natalie V. S. Vinkeles Melchers, Marielle Kloek, Luc E. Coffeng.

**Writing – original draft:** Natalie V. S. Vinkeles Melchers.

**Writing – review & editing:** Natalie V. S. Vinkeles Melchers, Wilma A. Stolk, Michele E. Murdoch, Belén Pedrique, Marielle Kloek, Roel Bakker, Sake J. de Vlas, Luc E. Coffeng.

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
