## [Decision Letter · Decision Letter 0]

18 Jan 2021

Dear Dr. Coffeng,

Thank you very much for submitting your manuscript "How does onchocerciasis-related skin and eye disease in Africa depend on cumulative exposure to infection and mass treatment?" for consideration at PLOS Neglected Tropical Diseases. As with all papers reviewed by the journal, your manuscript was reviewed by members of the editorial board and by several independent reviewers. The reviewers appreciated the attention to an important topic. Based on the reviews, we are likely to accept this manuscript for publication, providing that you modify the manuscript according to the review recommendations. 

Sincerely,

Keke C Fairfax, PhD

Deputy Editor

Keke Fairfax

Deputy Editor

Reviewer's Responses to Questions

**Key Review Criteria Required for Acceptance?**

**Methods**

-Are the objectives of the study clearly articulated with a clear testable hypothesis stated?

-Is the study design appropriate to address the stated objectives?

-Is the population clearly described and appropriate for the hypothesis being tested?

-Is the sample size sufficient to ensure adequate power to address the hypothesis being tested?

-Were correct statistical analysis used to support conclusions?

-Are there concerns about ethical or regulatory requirements being met?

Reviewer #1: While the methods are clearly stated and easy to follow, I do not have sufficient transmission modelling experience to comment on whether the models used are appropriate.

Reviewer #2: The paper submitted by Vinkeles Melchers et al. describes how new modules were developed within the general Onchosim model to represent the progression of clinical manifestations of onchocerciasis, and their regression after ivermectin treatment. Once the parameters defined, the authors applied the model to assess the long-term impact of repeated ivermectin treatments on the prevalence of the main cutaneous and ocular signs of the disease.

The work to reach these objectives is very impressive and all the methods used are very well described in the paper and in the detailed supplementary material. This is a major strength of this paper.

However, the authors certainly did not use all the published or potentially available data to “feed” the model and refine the various parameters. The parameters used to describe the relationships between the pre-treatment prevalence of infection and the prevalence of the various manifestations were evaluated using data from several thousands of people but those used for the cutaneous signs come from only four articles. One understands that the authors wished to include only papers where the cutaneous signs are defined according to Murdoch et al. (1993). Notwithstanding, they could have performed sensitivity analyses using data published in other articles which are not cited in this manuscript: for example, Edungbola et al., 1987, Carme et al., 1993, or Kollo et al., 1995, for depigmentation. What is the best approach to develop a model? relatively few precise data, or more less precise data? This has to be discussed.

Regarding the effects of ivermectin on cutaneous signs of onchocerciasis, the authors used data from about 350 subjects examined before and one year after a single dose of ivermectin (Brieger et al.) and about 5200 subjects examined before and 5-6 years after implementation of annual treatments (as part of a study funded by APOC). Even if the data on the effects of ivermectin on onchodermatitis is scarce, it is not the case for nodule prevalences, whose change over time has been published in a number of papers. Again, the authors should explain why they excluded these datasets.

Reviewer #3: The objectives of the study are clearly articulated, and the study design is appropriate to address the stated objectives. The overall methodology seems sound although additional details of the mathematical formulations used in calculating tissue damage would be appreciated, for fully assessing the approach. I don't have any concerns about ethical or regulatory requirements.

Other specific comments on the Methods section:

Line 147: Why are mf killed through treatment considered to not cause tissue damage? Is their contribution to morbidity considered too transitory? This should be justified and referenced.

Lines 154–160, 183-184: Similarly, these modeling decisions, while reasonable, should be justified either on the grounds that they provide flexibility in the model (i.e., require fewer constraining assumptions) or documentation of such variability from the literature.

Line 133: Please explain here how tissue damage is quantified, i.e., is it a unitless quantity whose meaning is only relevant within this model, or does it have a broader interpretation?

Line 293: Is this value of 0.5% prevalence consistent with programmatic goals as stated by WHO or individual country programs? If so, please cite. If not, please justify this choice.

**Results**

-Does the analysis presented match the analysis plan?

-Are the results clearly and completely presented?

-Are the figures (Tables, Images) of sufficient quality for clarity?

Reviewer #1: (No Response)

Reviewer #2: The analysis presented matches the analysis plan

The results are clearly and completely presented

The figures (Tables, Images) are of sufficient quality for clarity

Reviewer #3: The outline of the results matches the methods and the results are clearly and completely presented. The figures are useful and well-done, with minor suggestions below. My overarching suggestions for improvements are (1) to provide quantifications of in-sample and out-of-sample prediction error (currently only qualitative assessments of validation are presented), and (2) to incorporate measures of model uncertainty into the reporting of results, including figures (uncertainty is mentioned on lines 422–423 but otherwise is not addressed). These considerations are crucial to evaluating the robustness of the model estimates.

Other comments regarding the results:

Line 326: I believe that the threshold values for savanna areas is reversed and should read “…(1.7 vs. 3.1 thousand, respectively)…” Also, please discuss the relevance of focusing on relative differences in disease thresholds versus absolute differences, as the absolute differences are actually larger in forest areas.

Figure 4: Please clarify what the three sets of estimates for each clinical manifestation represent (not the three MDA coverage levels indicated by color); presumably these reflect hypo-, meso- and hyperendemic starting conditions? This should be stated in the figure caption, at least.

**Conclusions**

-Are the conclusions supported by the data presented?

-Are the limitations of analysis clearly described?

-Do the authors discuss how these data can be helpful to advance our understanding of the topic under study?

-Is public health relevance addressed?

Reviewer #1: (No Response)

Reviewer #2: As mentioned above, the authors should clearly explain why they used data from a relatively short list of studies, which could be regarded as a limitation of the paper, and why others were excluded, even for sensitivity analyses.

Regarding the data on the cutaneous manifestations, figure 3 shows that their prevalence can vary widely for a given prevalence of infection. The authors hypothesize that this variability could be due to differences in pathogenicity between savanna and forest “strains”. They could cite the classical papers by Anderson et al. (TRSTMH 1974), where this difference was explored, to strengthen this hypothesis.

The possibility that visual impairment could be reversible is considered in the Discussion section (from lines 512), and this is certainly a key point. More references on this possibility could be cited (for example Mabey et al., 1996; Chippaux et al., 1999; Banla et al., 2014 …).

Reviewer #3: The conclusions are generally supported by the data presented. The discussion sufficiently addresses limitations of the analysis, although I would like to see some mention of the limitations of the chosen modeling approach to quantifying uncertainty in the underlying parameters and estimates (e.g., as compared with a Bayesian statistical model). The authors clearly discuss the public health relevance of this work. I would like to see some discussion of any existing targets for morbidity reduction, such as from WHO or individual country programs. This would be valuable in understanding the context of the 0.5% prevalence target considered negligible by the authors.

Other comments:

Lines 458–459: Please provide the results supporting this conclusion in a table in the main text.

Line 507: Maybe replace “…related to the the biology of different parasite species” with something like “…related to genetic variation in O. volvulus.” 

Line 545: Please define DALYs.

**Editorial and Data Presentation Modifications?**

Reviewer #1: (No Response)

Reviewer #2: - S1, page 14, below the formula: remplace one “y_obs_frac” by “y_hat_frac”

- some references are incomplete (16, 64)

- The acknowledgments are restricted to the investigators of the Murdoch et al studies, and could be extended to the other investigators of the key papers used to develop the model, and to the organizations that funded these studies (TDR, APOC, …)

Reviewer #3: Comments on the Background:

Line 88: The introduction is excellent. I suggest only adding a very brief description of past estimates of onchocerciasis burden to the end of the first paragraph (line 88) to provide context for the scale of infection and morbidity prevalence, e.g., referencing the following sources:

 O’Hanlon, Simon J., Hannah C. Slater, Robert A. Cheke, Boakye A. Boatin, Luc E. Coffeng, Sébastien D. S. Pion, Michel Boussinesq, Honorat G. M. Zouré, Wilma A. Stolk, and María-Gloria Basáñez. “Model-Based Geostatistical Mapping of the Prevalence of Onchocerca Volvulus in West Africa.” PLOS Neglected Tropical Diseases 10, no. 1 (January 15, 2016): e0004328. https://doi.org/10.1371/journal.pntd.0004328.

 Kelly-Hope, Louise A., Janet Hemingway, Mark J. Taylor, and David H. Molyneux. “Increasing Evidence of Low Lymphatic Filariasis Prevalence in High Risk Loa Loa Areas in Central and West Africa: A Literature Review.” Parasites & Vectors 11, no. 1 (June 15, 2018): 349. https://doi.org/10.1186/s13071-018-2900-y.

 Zouré, Honorat GM, Mounkaila Noma, Afework H Tekle, Uche V Amazigo, Peter J Diggle, Emanuele Giorgi, and Jan HF Remme. “The Geographic Distribution of Onchocerciasis in the 20 Participating Countries of the African Programme for Onchocerciasis Control: (2) Pre-Control Endemicity Levels and Estimated Number Infected.” Parasites & Vectors 7, no. 1 (2014): 326. https://doi.org/10.1186/1756-3305-7-326.

 Sauerbrey, Mauricio, Lindsay J Rakers, and Frank O Richards. “Progress toward Elimination of Onchocerciasis in the Americas.” International Health 10, no. suppl_1 (March 1, 2018): i71–78. https://doi.org/10.1093/inthealth/ihx039.

Comments on the Supplementary Information:

Page 32: It is claimed that Figure S5 “shows that the model predictions fit quite well with available external data,” but my impression from the figure is that the fit is often actually not very good except for “any DPM” (recognizing that the panel for any vs. severe itch is not comparing identical quantities). Can model fit be quantified and reported, with perhaps a more nuanced discussion of model prediction accuracy?

Figures S11–S13: These are valuable figures, as they illustrate one important source of uncertainty in model estimates. However, the figures could be improved by either (1) reducing the opacity of each line, so that the central trends become more apparent, or (2) showing the median values with 95% UI displayed as a faint ribbon.

**Summary and General Comments**

Reviewer #1: (No Response)

Reviewer #2: A weakness of this paper is probably that there is only one clinician, and no ophthalmologist, among the authors. Clinicians would have probably helped in the development of some parts of the model. However, the paper is very interesting because it shows for the first time how the prevalence of each manifestation of onchocerciasis will tend to decrease in the future. However, as the parameters were assessed using data from relatively low numbers of subjects, they will probably evolve if more data becomes available.

Reviewer #3: This manuscript represents an important advance in the development of modeling tools to support public health programs for the reduction of onchocerciasis morbidity. I applaud the authors for an exceptionally well-written and thorough paper. The broad scope of the sensitivity analyses is especially appreciated. My suggestions for changes, as detailed in other portions of this review, are relatively minor and mainly serve to provide additional context for the analysis and to provide additional information with which to ascertain model predictive accuracy and uncertainty.

PLOS authors have the option to publish the peer review history of their article (what does this mean?). If published, this will include your full peer review and any attached files.

Reviewer #1: No

Reviewer #2: No

Reviewer #3: No
---

## [Editor Report · Decision Letter 1]

19 May 2021

Dear Dr. Coffeng,

We are pleased to inform you that your manuscript 'How does onchocerciasis-related skin and eye disease in Africa depend on cumulative exposure to infection and mass treatment?' has been provisionally accepted for publication in PLOS Neglected Tropical Diseases.

Best regards,

Keke C Fairfax, PhD

Deputy Editor

Keke Fairfax

Deputy Editor

---

## [Editor Report · Acceptance letter]

7 Jun 2021

Dear Dr. Coffeng,

We are delighted to inform you that your manuscript, "How does onchocerciasis-related skin and eye disease in Africa depend on cumulative exposure to infection and mass treatment?," has been formally accepted for publication in PLOS Neglected Tropical Diseases.

Best regards,

Shaden Kamhawi

co-Editor-in-Chief

Paul Brindley

co-Editor-in-Chief
